# Prevalence and Outcomes Associated with Vitamin D Deficiency among Indexed Hospitalizations with Cardiovascular Disease and Cerebrovascular Disorder—A Nationwide Study

**DOI:** 10.3390/medicines7110072

**Published:** 2020-11-22

**Authors:** Urvish Patel, Salma Yousuf, Komal Lakhani, Payu Raval, Nirmaljot Kaur, Toochukwu Okafor, Chail Shah, Harmandeep Singh, Mehwish Martin, Chika Nwodika, Angelina Yogarajah, Jigisha Rakholiya, Maitree Patel, Raja Chandra Chakinala, Shamik Shah

**Affiliations:** 1Department of Public Health, Icahn School of Medicine at Mount Sinai, New York, NY 10029, USA; dr.salmayousuf@gmail.com (S.Y.); mahwishmartin@gmail.com (M.M.); 2Department of Internal Medicine, Lenox Hill Hospital, Northwell Health, New York, NY 10075, USA; klakhani7684@gmail.com; 3siParadigm Diagnostic Informatics, Pine Brook, NJ 07058, USA; sheth.payu@gmail.com; 4Pramukhswami Medical College, Shree Krishna Hospital, Anand, Gujarat 388325, India; 5Department of Internal Medicine, Sri Guru Ram Das Institute of Medical Sciences and Research, Amritsar, Punjab 143501, India; nirmaljot94@gmail.com (N.K.); harman96@gmail.com (H.S.); 6Department of Internal Medicine, Larkin Community Hospital, Hialeah, FL 33012, USA; drlilianokafor@gmail.com; 7Department of Internal Medicine, Icahn School of Medicine at Mount Sinai, New York, NY 10029, USA; Chail.shah@yahoo.com; 8Brooklyn Cancer Care, Brooklyn, NY 11203, USA; 9Department of Internal Medicine, Oba Okunade Sijuade College of Medicine, Igbinedion University Okada, Edo State 23401, Nigeria; drchika.nwodika@gmail.com; 10Medical University of Americas, Devens, MA 01434, USA; angelinayogarajah@gmail.com; 11Department of Internal Medicine, Mayo Clinic, Rochester, MN 55905, USA; jigirakholiya@gmail.com; 12Department of Internal Medicine, Advent Health Orlando, Orlando, FL 32803, USA; maitree1989@gmail.com; 13Department of Internal Medicine, Guthrie Robert Packer hospital, Sayre, PA 18840, USA; rajachandra@gmail.com; 14Department of Neurology, Stormont Vail Health, Topeka, KS 66604, USA; drshahshamik@gmail.com

**Keywords:** vitamin D deficiency, ischemic heart disease, atrial fibrillation, congestive heart failure, myocardial infarction, ischemic stroke, hemorrhagic stroke, transient ischemic attack, nationwide inpatient sample

## Abstract

**Background**: According to past studies, recovery and survival following severe vascular events such as acute myocardial infarction and stroke are negatively impacted by vitamin D deficiency. However, the national estimate on disability-related burden is unclear. We intend to evaluate the prevalence and outcomes of vitamin D deficiency (VDD) among patients with cardiovascular disease (CVD) and cerebrovascular disorder (CeVD). **Methods**: We performed a cross-sectional study on the Nationwide Inpatient Sample data (2016–2017) of adult (≥18 years) hospitalizations. We identified patients with a secondary diagnosis of VDD and a primary diagnosis of CVD and CeVD using the 9th revision of the International Classification of Diseases, clinical modification code (ICD-10-CM) codes. A univariate and mixed-effect multivariable survey logistic regression analysis was performed to evaluate the prevalence, disability, and discharge disposition of patients with CVD and CeVD in the presence of VDD. **Results**: Among 58,259,589 USA hospitalizations, 3.44%, 2.15%, 0.06%, 1.28%, 11.49%, 1.71%, 0.38%, 0.23%, and 0.08% had primary admission of IHD, acute MI, angina, AFib, CHF, AIS, TIA, ICeH, and SAH, respectively and 1.82% had VDD. The prevalence of hospitalizations due to CHF (14.66% vs. 11.43%), AIS (1.87% vs. 1.71%), and TIA (0.4% vs. 0.38%) was higher among VDD patients as compared with non-VDD patients (*p* < 0.0001). In a regression analysis, as compare with non-VDD patients, the VDD patients were associated with higher odds of discharge to non-home facilities with an admission diagnosis of CHF (aOR 1.08, 95% CI 1.07–1.09), IHD (aOR 1.24, 95% CI 1.21–1.28), acute MI (aOR 1.23, 95% CI 1.19–1.28), AFib (aOR 1.21, 95% CI 1.16–1.27), and TIA (aOR 1.19, 95% CI 1.11–1.28). VDD was associated with higher odds of severe or extreme disability among patients hospitalized with AIS (aOR 1.1, 95% CI 1.06–1.14), ICeH (aOR 1.22, 95% CI 1.08–1.38), TIA (aOR 1.36, 95% CI 1.25–1.47), IHD (aOR 1.37, 95% CI 1.33–1.41), acute MI (aOR 1.44, 95% CI 1.38–1.49), AFib (aOR 1.10, 95% CI 1.06–1.15), and CHF (aOR 1.03, 95% CI 1.02–1.05) as compared with non-VDD. **Conclusions**: CVD and CeVD in the presence of VDD increase the disability and discharge to non-home facilities among USA hospitalizations. Future studies should be planned to evaluate the effect of VDD replacement for improving outcomes.

## 1. Introduction

Cardiovascular disease (CVD) and cerebrovascular disease (CeVD) together contribute to a major disease burden on the USA healthcare system. The National Vital Statistics Reports for 2017 indicate that CVD and CeVD are two of the 10 leading causes of total deaths in the United States [1]. Approximately 647,000 Americans die from heart disease each year and one in four deaths in the USA is because of cardiovascular disease [2,3]. The United States has spent about USD 219 billion each year from 2014 to 2015 towards the costs of heart disease including health care services, medicines, and the loss of productivity due to cardiovascular disease burden [3]. Herrick et al. evaluated that, from 2011 to 2014, approximately 5% and 18% of the USA population aged ≥ 1 year were at the risk of vitamin D deficiency and insufficiency, respectively [4]. Vitamin D deficiency was associated with 57% higher risk for prevalent CVD [5]. A meta-analysis by Zhang et al. concluded that there was no significant reduction in the blood pressure by supplementing vitamin D in hypertensive patients [6]. A prospective study consisting of 814 acute coronary syndrome patients by De Metrio et al. demonstrated that 19% and 70% of their sample size had vitamin D insufficiency and deficiency, respectively. Clinical investigations have inferred an association between hypovitaminosis D and acute myocardial infarction (AMI) along with increased morbidity and mortality in this clinical setting [7]. We know that atherosclerosis triggers myocardial infarction. Rahman et al. demonstrated that matrix metalloproteinases, which are proteins that contribute to aberrant cardiomyocyte remodeling in response to injury and atherosclerosis, were upregulated in vitamin D receptor knockout mice [8]. Lower levels of 25-hydroxycholecalciferol and calcitriol were found in patients with CHF, and 25-hydroxycholecalciferol levels > 40 ng/mL were required to decrease NT-proANP levels [9]. Results of a study by Vitezova et al. showed that there was no significant association between vitamin D levels and atrial fibrillation [10], whereas a study by Liu et al. reported that higher levels of vitamin D decreased the risk of atrial fibrillation in an older age group [11].

Recently, several studies have been performed to identify a relationship between vitamin D deficiency and its impact on cerebrovascular diseases. Lower vitamin D levels were not associated with higher stroke risk but were instead a consequence of stroke [12], rather, only severe vitamin D deficiency showed increased ischemic stroke risk [13]. A prospective cohort study, conducted in China, showed that vitamin D deficient patients were at a higher risk of exhibiting post-stroke cognitive impairment than those with vitamin D insufficient and vitamin D sufficient patients [14]. Fahmy et al. concluded that the likelihood of stroke increased > 13 times in a vitamin deficient patient [15]. Neuroimaging of patients with low 25-hydroxycholecalciferol revealed lacunes, severe white matter hyperintensity, and cerebral microbleeds in basal ganglia, thalamus, and brain stem, suggesting that chronic brain injury was associated with small vessel disease [16]. The severity of vitamin D deficiency appears to be a strong negative predictor for survival after ischemic stroke. A study showed that it was necessary to treat five patients with severe vitamin D deficiency to prevent one death [12].

The primary aim of this study was to evaluate the prevalence of cardiovascular disease (angina, myocardial infarction, arrhythmia, and congestive heart failure) and cerebrovascular disorder (minor stroke and ischemic stroke) among patients with vitamin D deficiency. The secondary aim was to find out whether cardiovascular and cerebrovascular diseases in the presence of vitamin D deficiency worsen discharge outcomes.

## 2. Materials and Methods

We gathered the National Inpatient Sample data from January 2016 to December 2017 from the Agency for Healthcare Research and Quality’s Healthcare Cost and Utilization Project (HCUP). It is the largest publicly available all-payer inpatient care database in the United States and contains discharge-level data provided by states that participate in the HCUP. This administrative dataset contains data on approximately 8 million hospitalizations in 1000 hospitals that are chosen to approximate a 20% stratified sample of all the United States’ community hospitals, representing more than 95% of the national inpatient admissions. Detailed information on NIS is available at http://www.hcup-us.ahrq.gov/db/nation/nis/nisdde.jsp.

### 2.1. Study Population

We used the 9th revision of the International Classification of Diseases, clinical modification code (ICD-10-CM) to identify adult patients admitted to hospital within two years (2016–2017) with a secondary diagnosis of vitamin D deficiency (ICD-10-CM code 577.8). Among those USA admissions, primary admissions with CVD (IHD, acute MI, angina, AFib, CHF) and CeVD (AIS, TIA, ICeH, and SAH) were identified using ICD-10-CM codes. We used ICD-10-CM codes to identify independent predictors (covariates), including the comorbidities of hypertension, diabetes mellitus, obesity, hypercholesterolemia, smoker, and renal dysfunction (end-stage renal disease, and chronic kidney disease) (Appendix A). Ages < 18 years and admissions with missing data for age, gender, and race were excluded. The sample size was based on the available data.

### 2.2. Patient and Hospital Characteristics

Patient characteristics of interest were sex, age, race, insurance status, and concomitant diagnoses, as mentioned above. The race was defined by White (referent), African American, Hispanic, Asian or Pacific Islander, and Native American. Insurance status was defined by Medicare (referent), Medicaid, private Insurance, and other/self-pay/no charge. We defined the severity of comorbid conditions using the Elixhauser comorbidity index [17] (Appendix A). A facility was considered to be a teaching hospital if they had an American Medical Association-approved residency program, was a member of the Council of Teaching Hospitals or had full-time equivalent interns and residents at a patient ratio of ≥ 0.25. The HCUP NIS contains data on total charges for each hospital in the databases, which represents the amount that hospitals billed for services.

### 2.3. Outcomes

The primary aim of this study was to evaluate the prevalence of CVD and CeVD among vitamin D deficiency patients. Diagnoses of vitamin D deficiency were based on ICD-10 codes rather than a lab-based diagnosis. The secondary aim was to evaluate the outcomes such as disability (loss of function) and discharge disposition associated with vascular events (index hospitalization/primary reason) among vitamin D deficiency as compared with no deficiency. The comparison of disability/loss of function was investigated by All-Patient Refined Diagnosis Related Groups (APR-DRGs) severity. The APR-DRGs severity was assigned using software developed by 3M Health Information Systems, where a score 1 indicates minor loss of function; 2, moderate; 3, major; and 4, extreme loss of function. Discharge disposition was defined by discharge to home vs. non-home (short-term hospital, skilled nursing facility, or intermediate care facility). Details of APR-DRGs can be found at https://www.hcup-us.ahrq.gov/db/vars/aprdrg_severity/nisnote.jsp.

### 2.4. Statistical Analysis

All statistical analyses were performed using the weighted survey methods in SAS (version 9.4). The *p*-values of < 0.05 were considered to be significant. The univariate analysis of differences between categorical variables was tested using the Chi-square test, and analysis of differences between continuous variables (age) was tested using paired Student’s *t*-test. Mixed-effects survey logistic regression models with weighted analysis were used to evaluate the effect modification of vitamin D deficiency on estimating outcomes with primary CVD and CeVD. We included demographics (age, gender, race), patient-level hospitalization variables (admission day, primary payer, admission type, median household income category), hospital-level variables (hospital region, teaching vs. non-teaching hospital, hospital bed size), comorbidities (hypertension, diabetes mellitus, obesity, hypercholesterolemia, smoker, and renal dysfunction (end-stage renal disease, and chronic kidney disease)), and Elixhauser comorbidity index. The goodness of fit of the model was evaluated by the c-value.

## 3. Results

### 3.1. Disease Hospitalizations

We found a total of 201,602,017 of USA hospitalizations from 2016 to 2017, out of which 1,059,755 (1.82%) had a secondary diagnosis of vitamin D deficiency after excluding patients with missing data for age, race, sex, and outcomes.

### 3.2. Demographics, Patient and Hospital Characteristics, and Comorbidities of Patients with Vitamin D Deficiency

Older patients (64 vs. 58 years), female (1.99% vs. 1.58% (male)), African American (2.1% vs. 1.85% (White) vs. 1.35% (Hispanic) vs. 1.67% (Asian) vs. 1.74% (Native-American), *p* < 0.0001), patients using Medicare (2.29% vs. 1.28% (Medicaid) vs. 1.56% (private insurance) vs. 1.05% (other/self-pay/no charge), *p* < 0.0001) and admitted in the Midwest region (2.40% vs. 1.90% (Northeast) vs. 1.57% (South) vs. 1.60% (West), *p* < 0.0001) had higher prevalence of vitamin D deficiency as compared with their counterparts (*p* < 0.0001). The prevalence of VDD was higher among females (63.42% vs. 57.80%, *p* < 0.0001), African American (18.00% vs. 15.60%, *p* < 0.0001), and patients on Medicare (59.89% vs. 47.26%, *p* < 0.0001) as compared with non-VDD. The prevalence of VDD was higher among diabetes mellitus (34.80% vs. 26.38%, *p* < 0.0001), hypertension (69.21% vs. 55.04%, *p* < 0.0001), obesity (24.08% vs. 15.12%, *p* < 0.0001), hyperlipidemia (49.96% vs. 31.65%, *p* < 0.0001), and renal failure (chronic and ESRD) (34.54% vs. 23.66%, *p* < 0.0001) as compared with non-VDD (Table 1).

### 3.3. Prevalence of Vascular Events

The prevalence of cardiovascular diseases such as ischemic heart disease, myocardial infarction, angina, atrial fibrillation, and congestive heart failure was 3.44%, 2.15%, 0.06%, 1.28%, and 11.49%, respectively. The prevalence of cerebrovascular disorders such as acute ischemic stroke, transient ischemic attack, intracerebral hemorrhage, subarachnoid hemorrhage was 1.71%, 0.38%, 0.23% and 0.08%, respectively.

The prevalence of different cardio- and cerebrovascular events (primary cause of/index hospitalizations) among VDD patients are listed in Table 2. Among patients admitted with different vascular events, the prevalence of congestive heart failure (14.66% vs. 11.43%, *p* < 0.0001), acute ischemic stroke (1.87% vs. 1.71%, *p* < 0.0001), and transient ischemic attack (0.40% vs. 0.38%, *p* = 0.0014) were higher among VDD as compared with non-VDD.

### 3.4. Outcomes

#### 3.4.1. Univariable Outcomes

Table 3 shows the univariable outcomes of cardiovascular and cerebrovascular events in VDD. The patients with VDD had a higher prevalence of severe or extreme disability (45.70% vs. 34.79%, *p* < 0.0001) and discharged non-home (44.25% vs. 28.82%, *p* < 0.0001) as compared with non-vitamin D deficiency.

The prevalence of severe or extreme disability was higher among VDD patients admitted with vascular events (ischemic heart disease (52.92% vs. 38.44%, *p* < 0.0001), acute myocardial ischemia (56.88% vs. 40.29%, *p* < 0.0001), angina (17.59% vs. 11.94%, *p* < 0.0001), atrial fibrillation (40.28% vs. 32.86%, *p* < 0.0001), congestive heart failure (76.08% vs. 72.74%, *p* < 0.0001), acute ischemic stroke (46.54% vs. 40.93%, *p* < 0.0001), transient ischemic attack (27.52% vs. 18.77%, *p* < 0.0001), intracerebral hemorrhage (72.12% vs. 65.03%, *p* < 0.0001), and subarachnoid hemorrhage (93.81% vs. 91.07%, *p* = 0.0346)) as compared with patients without VDD and vascular events.

Vitamin D deficient patients had a higher prevalence of discharged non-home during vascular events (ischemic heart disease (41.74% vs. 31.95%, *p* < 0.0001), acute myocardial infarction (43.61% vs. 32.81%, *p* < 0.0001), angina (22.52% vs. 18.58%, *p* = 0.0182), atrial fibrillation (31.63% vs. 24.17%, *p* < 0.0001), congestive heart failure (61.72% vs. 58.45%, *p* < 0.0001), acute ischemic stroke (66.56% vs. 62.32%, *p* < 0.0001), transient ischemic attack (35.15% vs. 28.99%, *p* < 0.0001), and intracerebral hemorrhage (81.12% vs. 76.03%, *p* < 0.0001)) as compared with patients without-VDD and vascular events.

#### 3.4.2. Regression Model Derivation for APR-DRG Severity (Severe or Extreme Disability)

Table 4 also shows the different models and a comparison of the status of VDD and severe or extreme disability, considering vascular events that occurred in case (VDD) and control (non-VDD). As compared with non-VDD, the VDD patients were associated with higher odds of severe or extreme disability in patients primarily admitted with congestive heart failure [adjusted odds ratio (aOR) 1.03, 95% confidence interval (95% CI) 1.02–1.05, *p* < 0.0001], ischemic heart disease (aOR 1.37, 95% CI 1.33–1.41, *p* < 0.0001), myocardial infarction (aOR 1.44, 95% CI 1.38–1.49, *p* < 0.0001), atrial fibrillation (aOR 1.10, 95% CI 1.06–1.15, *p* < 0.0001), and acute ischemic stroke (aOR 1.10, 95% CI 1.06–1.14, *p* < 0.0001), transient ischemic stroke (aOR 1.36, 95% CI 1.25–1.47, *p* < 0.0001), and intracerebral hemorrhage (aOR 1.22, 95% CI 1.08–1.38, *p* < 0.0001).

#### 3.4.3. Regression Model Derivation for Discharge to Non-Home

In Table 4, we describe the different models, comparing the status of VDD and discharge disposition to non-home vs. home, considering vascular events occurred in case (VDD) and control (non-VDD) and after adjusting models for patients’ demographics, patient and hospital level characteristics, comorbidities, and Elixhauser comorbidity index.

As compared with non-VDD, the VDD patients were associated with a higher odds of discharge to non-home among patients primarily admitted with congestive heart failure (aOR 1.08, 95% CI 1.07–1.09, *p* < 0.0001), ischemic heart disease (aOR 1.24, 95% CI 1.21–1.28, *p* < 0.0001), myocardial infarction (aOR 1.23, 95% CI 1.19–1.28, *p* < 0.0001), atrial fibrillation (aOR 1.21, 95% CI 1.16–1.27, *p* < 0.0001), and transient ischemic stroke (aOR 1.19, 95% CI 1.11–1.28, *p* < 0.0001).

## 4. Discussion

The national inpatient data were analyzed to look for the demographics and the outcomes of vascular events (i.e., cardiovascular and cerebrovascular events) in patients with vitamin D deficiency. Along with the prevalence of vascular events, we assessed the prevalence of disability and discharge disposition in these patients. A study that collected data through the National Health and Nutrition Examination Survey (NHANES) from 2011 to 2012 showed that 39.92% of the population had vitamin D deficiency; similar results were seen when data were collected from 2005 to 2006 by NHANES [18]. In our study, the prevalence of vitamin D deficiency was more in females and African populations as compared with males and other races. Similar results were noted in a study conducted by Scragg et al. [19]. Another study by Luijtgaarden et al. found that the total prevalence of vitamin D deficiency was 45% [20]. Our study reported 57.90% females affected by vitamin D deficiency which was almost similar to another study [21] which reported 62.70% females affected.

We noted that hypertension was common in vitamin D deficient patients, whereas, in a study by Scragg et al., it was mentioned that serum 25-OH-D levels were not affected whether the patients were hypertensive or non-hypertensive [19]. This was because vitamin D decreased the renin-angiotensin-aldosterone system pathway, thus affecting the systolic blood pressure. Lee et al. confirmed the high prevalence of vitamin D deficiency seen in myocardial infarction patients [22]. However, our study showed that vitamin D deficiency was lower in myocardial infarction patients. An inverse correlation was seen between serum vitamin D levels and congestive heart failure. Several other studies have manifested similar results [23,24]. Scragg et al. conducted a randomized control trial and reported that monthly supplements of vitamin D did not prevent cardiovascular disease [25]. The prevalence of vitamin D deficiency was lower in ischemic heart disease, similar to our study. Contrarily, Wasson et al. demonstrated that marked vitamin D deficiency was related to increased risk of ischemic heart disease [26]. Aleksova et al. conducted a single center study and reported an independent association of low vitamin D levels with poor outcomes of ACS in hospital and at one-year follow-up [27].

The relationship of vitamin D deficiency to cardiovascular diseases is multipronged. The association of vitamin D with coronary heart diseases could be via blood pressure, glycemic control, or parathyroid hormone (PTH). Vitamin D deficiency causes an increase in the circulating PTH level that, in turn, is known to promote atherosclerosis [28]. Vitamin D deficiency also affects people with type 2 diabetes mellitus by its effect on pancreatic β-cell function and insulin resistance [29]. In addition, Vitamin D downregulates proinflammatory cytokines (TNF-α and IL-6), upregulates anti-inflammatory cytokine (IL-10), inhibits vascular smooth muscle proliferation, suppresses vascular calcification, and acts as a negative endocrine regulator of the renin-angiotensin system [30]. Another study by Monica De Metrio suggested that patients with lower vitamin D levels had higher high-sensitivity C-reactive protein (CRP) values, which suggested a possible link between low vitamin D levels and inflammation [31]. However, the relationship between them is controversial, as recently reported by Eren et al. who found that there was no association among vitamin D levels and inflammatory markers in ACS [32].

An animal study by Neveu et al. concluded that the synthesis of nerve growth factor was directly associated with vitamin D levels [33]. Therefore, it is clear that vitamin D should be adequate to protect nerve cells. Our results show that VDD increases the risk of acute ischemic stroke. In addition, it was noted that there were poor outcomes of acute ischemic stroke in patients with VDD by Alfieri et al. and Chen et al. [14,34]. There is a lack of data on how VDD affects transient ischemic attack, but in our study, the prevalence of transient ischemic attack is higher in VDD as compared with non-VDD.

Zittermann et al. (EVITA study) showed that 4000 IU of vitamin D replacement had no association with mortality change in patients with heart disease and suggested caution for long-term supplementation with high vitamin D doses [35]. In a RCT of 25,871 participants, Manson et al. (VITAL study) mentioned a placebo effect for reducing the incidence of cardiovascular events following vitamin D replacement [36]. More prospective studies should be planned to evaluate the role of replacement therapy to reduce disability among patients with CeVD and CVD.

This study has several strengths including the large sample size with almost equal representation of male and female population, creating a normal distribution of the cohort with high statistical power of the study. Another highlight of this study is the positive associations among vitamin D deficiencies and in-hospital outcomes following vascular events among nationwide data representing more than 95% inpatient hospitalizations. Some of the limitations of our study are worth mentioning. Factors that affect vitamin D status, such as latitude, season, sunlight exposure, skin color, vitamin D intake, and serum albumin, etc., were not considered and may have influenced our results. We may have underestimated real vitamin D deficiency prevalence as the calculated prevalence was based on vascular events and in-hospital diagnosis. We may have missed patients with subclinical deficiency or deficiencies identified in out-patient settings. The diagnosis was based on ICD coding system, and hence was susceptible to coding errors. We used the APR-DRG coding system to evaluate disability and it is very reliable, externally validated and previously used by other studies [37,38].

## 5. Conclusions

The presence of vitamin D deficiency was associated with the risk of poor outcomes following CVD and CeVD even after adjusting for conventional risk factors. VDD patients were also associated with higher odds of discharge to non-home facilities and severe or extreme disability amongst cardiovascular diseases and cerebrovascular disorders as compared with those without VDD. There is a dire need to test this association in more cross-sectional and cohort studies. Population awareness of vitamin D deficiency must be created, and its possible cardiovascular and cerebrovascular harm should be alleviated by encouraging supplemental intake. Further investigation is also needed to understand whether supplementation of vitamin D in patients with CVD and CeVD reduces the incidence of complications and improves patients’ in-hospital outcomes.

## Figures and Tables

**Table 1 medicines-07-00072-t001:** Characteristics of USA hospitalizations identified with vitamin D deficiency.

Characteristics	Vitamin D Deficiency	Non-Vitamin D Deficiency	Total	*p*-Value
Unweighted (%)	211,951 (1.82)	11,439,974 (98.18)	11,651,925 (100)	
Weighted (%)	1,059,755 (1.82)	57,199,834 (98.18)	58,259,589 (100)	<0.0001
**Demographics of Patients**				
**Age Mean (SD) (Years)**	64 (18)	58 (20)		<0.0001
**Gender (%)**				<0.0001
Male	36.58	42.20	42.10	
Female	63.42	57.80	57.90	
**Race (%)**				<0.0001
White	70.47	69.59	69.61	
African American	18.00	15.60	15.64	
Hispanic	8.35	11.36	11.30	
Asian or Pacific Islander	2.57	2.81	2.81	
Native American	0.61	0.64	0.64	
**Characteristics of Patients**				
**Median Household Income Category for Patient’s Zip Code (%)**				<0.0001
0–25th percentile	30.13	31.06	31.04	
26–50th percentile	25.67	25.96	25.96	
51–75th percentile	24.23	23.42	23.43	
76–100th percentile	19.98	19.56	19.57	
**Primary Payer (%)**				<0.0001
Medicare	59.89	47.26	47.49	
Medicaid	13.09	18.74	18.64	
Private insurance	22.91	26.85	26.78	
Other/self-pay/no charge	4.11	7.15	7.09	
**Admission Type (%)**				<0.0001
Non-elective	75.92	76.38	76.37	
Elective	24.08	23.62	23.63	
**Admission Day (%)**				<0.0001
Weekday	81.30	79.51	79.54	
Weekend	18.70	20.49	20.46	
**Characteristics of Hospitals**				
**Bed Size of Hospital (%) ***				<0.0001
Small	18.65	19.47	19.46	
Medium	27.95	29.43	29.40	
Large	53.40	51.10	51.14	
**Hospital Location and Teaching Status (%)**				<0.0001
Rural	8.47	9.09	9.08	
Urban non-teaching	20.62	24.94	24.86	
Urban teaching	70.91	65.97	66.06	
**Hospital Region (%)**				<0.0001
Northeast	20.17	19.28	19.30	
Midwest	28.39	21.41	21.54	
South	34.66	40.24	40.14	
West	16.78	19.07	19.03	
**Comorbidities/Confounders (%)**				
Diabetes mellitus	34.80	26.38	26.54	<0.0001
Hypertension	69.21	55.04	55.29	<0.0001
Obesity	24.08	15.12	15.28	<0.0001
Hyperlipidemia	49.96	31.65	31.98	<0.0001
Renal failure	34.54	23.66	23.85	<0.0001
Smoking	13.66	16.05	16.00	<0.0001

* Bed size of hospital indicates number of hospital beds which varies depending on hospital location (rural/urban), teaching status (teaching/non-teaching), and region (Northeast/Midwest/Southern/Western). Percentage in brackets in columns indicate direct comparison between vitamin D deficiency vs. non-vitamin D deficiency among USA index hospitalizations. Vitamin D deficiency was considered to be a secondary diagnosis and vascular events were taken to be primary diagnoses.

**Table 2 medicines-07-00072-t002:** Prevalence of cardiovascular diseases and cerebrovascular disorders among hospitalized patients with vitamin D deficiency.

Vascular Events	Vitamin D Deficiency	Non-Vitamin D Deficiency	Total	*p*-Value
Weighted (%)	1,059,755 (1.82)	57,199,834 (98.18)	58,259,589 (100)	<0.0001
**Cardiovascular Events (%)**				
Ischemic heart disease	2.62	3.45	3.44	<0.0001
Myocardial infarction	1.58	2.16	2.15	<0.0001
Angina	0.05	0.06	0.06	0.0093
Atrial fibrillation	1.14	1.28	1.28	<0.0001
Congestive heart failure	14.66	11.43	11.49	<0.0001
**Cerebrovascular Events (%)**				
Acute ischemic stroke	1.87	1.71	1.71	<0.0001
Transient ischemic attack	0.40	0.38	0.38	0.0014
Intracerebral hemorrhage	0.17	0.23	0.23	<0.0001
Subarachnoid hemorrhage	0.05	0.08	0.08	<0.0001

Percentages in brackets in columns indicate direct comparison between vitamin D deficiency vs. non-vitamin D deficiency among USA index hospitalizations. Vitamin D deficiency was considered to be a secondary diagnosis and vascular events were taken to be a primary diagnoses.

**Table 3 medicines-07-00072-t003:** Univariable outcomes of cardiovascular diseases and cerebrovascular disorders in the presence of vitamin D deficiency.

Outcomes	Vitamin D Deficiency	Non-Vitamin D Deficiency	Total	*p*-Value
Weighted (%)	1,059,755 (1.82)	57,199,834 (98.1)	58,259,589 (100)	<0.0001
**APR-DRG Severity/Disability (Loss of Function) (%) ^#^**				
Mild-moderate (overall)	54.30	65.21	65.01	<0.0001
Severe-extreme (overall)	45.70	34.79	34.99	
Mild-moderate (ischemic heart disease = 1)	47.08	61.56	61.36	<0.0001
Severe-extreme (ischemic heart disease = 1)	52.92	38.44	38.64	
Mild-moderate (acute myocardial ischemia = 1)	43.12	59.71	59.48	<0.0001
Severe-extreme (acute myocardial ischemia = 1)	56.88	40.29	40.52	
Mild-moderate (angina = 1)	82.41	88.06	87.97	<0.0001
Severe-extreme (angina = 1)	17.59	11.94	12.03	
Mild-moderate (atrial fibrillation = 1)	59.72	67.14	67.02	<0.0001
Severe-extreme (atrial fibrillation = 1)	40.28	32.86	32.98	
Mild-moderate (congestive heart failure = 1)	23.92	27.26	27.18	<0.0001
Severe-extreme (congestive heart failure = 1)	76.08	72.74	72.82	
Mild-moderate (acute ischemic stroke = 1)	53.46	59.07	58.95	<0.0001
Severe-extreme (acute ischemic stroke = 1)	46.54	40.93	41.05	
Mild-moderate (transient ischemic attack = 1)	72.48	81.23	81.06	<0.0001
Severe-extreme (transient ischemic attack = 1)	27.52	18.77	18.94	
Mild-moderate (intracerebral hemorrhage = 1)	27.88	34.97	34.86	<0.0001
Severe-extreme (intracerebral hemorrhage = 1)	72.12	65.03	65.14	
Mild-moderate (subarachnoid hemorrhage = 1)	6.19	8.93	8.90	0.0346
Severe-extreme (subarachnoid hemorrhage = 1)	93.81	91.07	91.10	
**Discharge Home vs. Non-Home ***				
Home (overall)	55.75	71.18	70.93	<0.0001
Non-home (SNF/SNF/HHC) (overall)	44.25	28.82	29.07	
Home (ischemic heart disease = 1)	58.26	68.05	67.91	<0.0001
Non-home (ischemic heart disease = 1)	41.74	31.95	32.09	
Home (Acute Myocardial Infarction = 1)	56.39	67.19	67.05	<0.0001
Non-home (acute myocardial infarction = 1)	43.61	32.81	32.95	
Home (angina = 1)	77.48	81.42	81.35	0.0182
Non-home (angina = 1)	22.52	18.58	18.65	
Home (atrial fibrillation = 1)	68.37	75.83	75.71	<0.0001
Non-home (atrial fibrillation = 1)	31.63	24.17	24.29	
Home (congestive heart failure = 1)	38.28	41.55	41.47	<0.0001
Non-home (congestive heart failure = 1)	61.72	58.45	58.53	
Home (acute ischemic stroke = 1)	33.44	37.68	37.59	<0.0001
Non-home (acute ischemic stroke = 1)	66.56	62.32	62.41	
Home (transient ischemic attack = 1)	64.85	71.01	70.89	<0.0001
Non-home (transient ischemic attack = 1)	35.15	28.99	29.11	
Home (intracerebral hemorrhage = 1)	18.88	23.97	23.89	<0.0001
Non-home (intracerebral hemorrhage = 1)	81.12	76.03	76.11	
Home (subarachnoid hemorrhage = 1)	42.86	44.59	44.57	0.4419
Non-home (subarachnoid hemorrhage = 1)	57.14	55.41	55.43	

^#^ APR-DRG severity/disability (loss of function) was assigned using software developed by 3M Health Information Systems, where a score of 1 indicates minor loss of function; 2, moderate; 3, major; 4, extreme loss of function. * Discharge non-home was defined as discharge to short-term hospital, skilled nursing facility, or intermediate care facility.

**Table 4 medicines-07-00072-t004:** Multivariable regression analysis of predictors of poor outcomes of cardiovascular diseases and cerebrovascular disorders among vitamin D deficiency patients.

Outcomes	Odds Ratio	Confidence Interval		*p*-Value	c-Value
		**LL**	**UL**		
**APR-DRG Disability (Severe and Extreme Loss of Function) ^#^**					
Non-vitamin D deficiency (congestive heart failure = 1)	Reference				
Vitamin D deficiency (congestive heart failure = 1)	1.03	1.02	1.05	<0.0001	0.835
Non-vitamin D deficiency (ischemic heart disease = 1)	Reference				
Vitamin D deficiency (ischemic heart disease = 1)	1.37	1.33	1.41	<0.0001	0.833
Non-vitamin D deficiency (acute myocardial infarction = 1)	Reference				
Vitamin D deficiency (acute myocardial infarction = 1)	1.44	1.38	1.49	<0.0001	0.833
Non-vitamin D deficiency (angina = 1)	Reference				
Vitamin D deficiency (angina = 1)	1.27	0.99	1.63	<0.0001	0.833
Non-vitamin D deficiency (atrial fibrillation = 1)	Reference				
Vitamin D deficiency (atrial fibrillation = 1)	1.10	1.06	1.15	<0.0001	0.834
Non-vitamin D deficiency (acute ischemic stroke = 1)	Reference				
Vitamin D deficiency (acute ischemic stroke = 1)	1.10	1.06	1.14	<0.0001	0.833
Non-vitamin D deficiency (transient ischemic attack = 1)	Reference				
Vitamin D deficiency (transient ischemic attack = 1)	1.36	1.25	1.47	<0.0001	0.834
Non-vitamin D deficiency (intracerebral hemorrhage = 1)	Reference				
Vitamin D deficiency (intracerebral hemorrhage = 1)	1.22	1.08	1.38	<0.0001	0.834
Non-vitamin D deficiency (subarachnoid hemorrhage = 1)	Reference				
Vitamin D deficiency (subarachnoid hemorrhage = 1)	1.35	0.92	1.99	<0.0001	0.834
**Discharge Non-Home ***					
Non-vitamin D deficiency (congestive heart failure = 1)	Reference				
Vitamin D deficiency (congestive heart failure = 1)	1.08	1.07	1.09	<0.0001	0.786
Non-vitamin D deficiency (Ischemic heart disease = 1)	Reference				
Vitamin D deficiency (Ischemic heart disease = 1)	1.24	1.21	1.28	<0.0001	0.786
Non-vitamin D deficiency (acute myocardial infarction = 1)	Reference				
Vitamin D deficiency (acute myocardial infarction = 1)	1.23	1.19	1.28	<0.0001	0.786
Non-vitamin D deficiency (angina = 1)	Reference				
Vitamin D deficiency (angina = 1)	1.15	0.92	1.42	<0.0001	0.785
Non-vitamin D deficiency (atrial fibrillation = 1)	Reference				
Vitamin D deficiency (atrial fibrillation = 1)	1.21	1.16	1.27	<0.0001	0.787
Non-vitamin D deficiency (acute ischemic stroke = 1)	Reference				
Vitamin D deficiency (acute ischemic stroke = 1)	1.03	1.00	1.06	<0.0001	0.787
Non-vitamin D deficiency (transient ischemic stroke = 1)	Reference				
Vitamin D deficiency (transient ischemic stroke = 1)	1.19	1.11	1.28	<0.0001	0.786
Non-vitamin D deficiency (intracerebral hemorrhage = 1)	Reference				
Vitamin D deficiency (intracerebral hemorrhage = 1)	1.06	0.92	1.21	<0.0001	0.786
Non-vitamin D deficiency (subarachnoid hemorrhage = 1)	Reference				
Vitamin D deficiency (subarachnoid hemorrhage = 1)	0.82	0.67	1.00	<0.0001	0.785

UL, Upper Limit; LL, Lower Limit. ^#^ APR-DRG severity/disability (loss of function) was assigned using software developed by 3M Health Information Systems, where a score of 1 indicates minor loss of function; 2, moderate; 3, major; and 4, extreme loss of function. * Discharge non-home was defined as discharge to short-term hospital, skilled nursing facility, or intermediate care facility. Models were adjusted for demographics (age, gender, race), patient-level hospitalization variables (admission day, primary payer, admission type, median household income category), hospital-level variables (hospital region, teaching vs. non-teaching hospital, hospital bed size), and comorbidities such as hypertension, diabetes mellitus, obesity, hypercholesterolemia, smoker, and renal dysfunction (end-stage renal disease and chronic kidney disease), and Elixhauser comorbidity index.

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
