# Peer review of "Prevalence and Outcomes Associated with Vitamin D Deficiency among Indexed Hospitalizations with Cardiovascular Disease and Cerebrovascular Disorder—A Nationwide Study"

_medicines, 2020, doi:10.3390/medicines7110072_

Round 1
Reviewer 1 Report
The study investigates a) the association between vitamin D deficiency (VDD) and major cardiovascular (CVD) as cerebrovascular disease (CeVD) among hospitalized patients for the same vascular disorders, b) association of VDD and discharge outcomes. Authors concludes that CVD and CVD in association with VDD increase the risk of post-discharge disability.
Although the topic sounds relevant, major revisions can ben considered prior to publication.
Methods and results
- Primary outcome is inconsistently described thorough the text (Introduction: "....prevalence of VDD amongst patients with CVD...", Methods: "...prevalence of CVD and CeVD amongst VDD..."). Table 1 and subsequent analysis stratify patients characteristics by VDD status. Authors may clarify this issue.
- Complete list of confounders, included in multivariate models, should be reported within statistical analysis section.
- Prevalence of VDD was very low (1.82%) (Table 1). Could prevalence be underestimated, secondary to retrospective diagnosis of VDD by ICD coding? This point is crucial and could be extensively discussed.
- Table 4: repeated lines with No-VDD as reference are not informative and could be removed
Introduction and discussion
- Association between VDD and post-discharge disability represents the most intriguing result of the study. More space should be dedicated to it in both introduction and discussion
- More extensive description of APR-DRG severity/disability could be helpful for readers
Conclusion
- Data poorly support that VDD was associated with risk of CVD and CeVD, because all included patients were affected by CVD or CeVD. Authors may clarify on it.
Author Response
Reviewer 1:
Comment 1: “Primary outcome is inconsistently described thorough the text (Introduction: "....prevalence of VDD amongst patients with CVD...", Methods: "...prevalence of CVD and CeVD amongst VDD..."). Table 1 and subsequent analysis stratify patients characteristics by VDD status. Authors may clarify this issue.”
We fix the last paragraph of the introduction “The primary aim of this study was to evaluate the prevalence of cardiovascular disease (angina, myocardial infarction, arrhythmia, and congestive heart failure) and cerebrovascular disorders (minor stroke and ischemic stroke) amongst patients with vitamin D deficiency.”
Comment 2: “Complete list of confounders, included in multivariate models, should be reported within statistical analysis section.”
We have added confounders in the analysis section as well beside the data collection section and tables
Comment 3: “Prevalence of VDD was very low (1.82%) (Table 1). Could prevalence be underestimated, secondary to retrospective diagnosis of VDD by ICD coding? This point is crucial and could be extensively discussed.”
We have now discussed all this explanation in limitation section of study. Please let us know if we need further explanations.
Comment 4: “Table 4: repeated lines with No-VDD as reference are not informative and could be removed”
We are keeping it as no-VDD in reference is different in each category for eg for MI, AIS, CHF, etc. Keeping it one time may create unnecessary confusion amongst readers.
Comment 5: Association between VDD and post-discharge disability represents the most intriguing result of the study. More space should be dedicated to it in both introduction and discussion
Due to word limitation, we try to balance discussion section between disability and discharge outcomes.
Comment 6: More extensive description of APR-DRG severity/disability could be helpful for readers
Described in the outcome section and the link is also added for references.
Comment 7: Data poorly support that VDD was associated with risk of CVD and CeVD, because all included patients were affected by CVD or CeVD. Authors may clarify on it.
Fixed “associated with risk of poor outcomes following CVD and CeVD even after adjusting for conventional risk factors”
Reviewer 2 Report
The study presents a cross-sectional analysis of the prevalence of low vitamin D status and to what extend is this condition associated with the risk of disability after discharge from hospital (discharge to non-home facility…).
The study was carried out in a methodologically appropriate manner and its results are undoubtedly very interesting. Being based on the national-wide registry, its sample size is very impressive. In general, I have no major objections and the analysis certainly deserves to be published.
I have only two minor remarks.
- From the point of view of possible clinical interpretation, the usual problem is the observational design of the study. It is difficult to decide whether the low status of vitamin D plays a direct pathophysiological role in the prognosis of patients or just reflected their poor functional status before admission (a hypothetical mechanism could be, for example, that less mobile patients are also less exposed to sunlight). Despite an unusually large sample size, from this point of view, the presented study does not bring anything fundamentally new to the current uncertain situation. Moreover, the results of the intervention studies do not give us much optimism in this regard (for example EVITA, VITAL…) I guess, that even this „pessimistic side of problem“ should be at least mentioned in the Conclusion.
- One of the most important problems of vitamin D status is its huge seasonality. So it would probably be good to adjust the results for at least the month when the hospitalization took place - I guess the investigators have this data available.
Author Response
Reviewer 2:
Comment 1: “From the point of view of possible clinical interpretation, the usual problem is the observational design of the study. It is difficult to decide whether the low status of vitamin D plays a direct pathophysiological role in the prognosis of patients or just reflected their poor functional status before admission (a hypothetical mechanism could be, for example, that less mobile patients are also less exposed to sunlight). Despite an unusually large sample size, from this point of view, the presented study does not bring anything fundamentally new to the current uncertain situation. Moreover, the results of the intervention studies do not give us much optimism in this regard (for example EVITA, VITAL…) I guess, that even this „pessimistic side of problem“ should be at least mentioned in the Conclusion.”
We have added findings from VITAL and EVITA trials.
Comment 2: “One of the most important problems of vitamin D status is its huge seasonality. So it would probably be good to adjust the results for at least the month when the hospitalization took place - I guess the investigators have this data available.”
We have month-wise data of admission but due to the following reasons, we have not adjusted and mentioned months-wise admission rates (1) 68464 have missing data on that which will be removed from the original model if a month is adjusted, (2) monthly admission rate amongst VDD and non-VDD are almost normally distributed between 7-9% without drastic change. New supplementary file with monthly distribution has attached for your evaluation purpose only.

Round 2
Reviewer 1 Report
The revised version of the manuscript improved. Howeverr, Authors' response to previous comments (suggestions from 4 to 7) were almost minimal. The paper could be suitable for publication whenever Editor will accept the position expressed by the Authors.